# Dex: array programming with typed indices

Dougal Maclaurin[1], Alexey Radul[1], Matthew J. Johnson[1], and Dimitrios Vytiniotis[2]

[1]Google Research
[2]DeepMind

## Abstract

Array programming is harder than it should be. Major pain points are managing bulk operations on high-rank arrays, and the associated shape and indexing errors. We describe Dex, a functional array processing language in the Haskell/ML family. Dex introduces a lightweight looping construct and a type system that captures common patterns of array shapes. We hope the language ideas we present here can influence the design of existing array programming systems.

## 1 Introduction

The Matlab/NumPy model of array programming avoids explicit loops and indexing and it can be impressively concise. But it can also feel obfuscated. For example, here's a NumPy function to compute the matrix of pairwise $L_1$ distances between the rows of a matrix (or is it the columns?).

```python
def pairwiseL1(x):
  return sum(abs(x.T - x[..., newaxis]), axis=1)
```

The traditional scalar loops of Fortran and C are more explicit, but much heavier. They require attention to the details of array initialization, iteration ranges, and order of mutating writes.

```cpp
void pairwiseL1(const vector<vector<float> > & x,
                vector<vector<float> > & out) {
  for (int i = 0; i < x.size(); i++) {
    for (int j = 0; j < x.size(); j++) {
      out[i][j] = 0;
      for (int k = 0; k < x[i].size(); k++) {
        out[i][j] += abs(x[i][k] - x[j][k]);}}}}
```

Can we have the best of both? The clarity of indexing explicitly and the succinctness of looping implicitly? Perhaps! In Dex (named after "index") we write `pairwiseL1` like this:

```
pairwiseL1 :: n=>d=>Real -> n=>n=>Real
pairwiseL1 x = for i j.
  sum (for k. abs (x.i.k - x.j.k))
```

The key innovation is to treat index sets as types rather than values, and use conventional type inference to infer iteration spaces. Leaning on inference leads to a lightweight and expressive looping syntax. In addition, all shape and indexing errors are static, and opportunities for parallelism are exposed to the compiler.

Our starting point is an analogy between arrays and functions.

## 2 Arrays are like functions

Mathematically, arrays are functions on finite index sets. In programming languages we distinguish arrays from functions, but not always by much! For example:

```
f(x, y)   # function application
x[i, j]   # array indexing
```

But where's the array equivalent of `lambda`? Let's create one. We'll call it an *index comprehension* and write it as `for i. <expr>`. It constructs an array whose elements are the result of evaluating `<expr>` at each index `i`. To illustrate, consider the problem of swapping the arguments of a two-argument function `f`. Here are three ways to write it in a Haskell-like language.

```
g = flip f          -- pointfree style
g = lam x y. f y x  -- pointed style
g x y = f y x       -- pointed syntactic sugar
```

The analogous problem for arrays is to transpose the axes of a 2-dimensional array x. Here are the three corresponding ways to write it in Dex.[1]

```
y = transpose x      -- pointfree style
y = for i j. x.j.i   -- pointed style
y.i.j = x.j.i        -- pointed syntactic sugar
```

The first version is how we transpose arrays in MAT-LAB/NumPy, using a built-in transpose function. The second version shows Dex's pointed alternative: the index comprehension. The third version, just syntactic sugar for the previous one, feels a lot like mathematical index notation. To finish the analogy, we introduce the type of arrays a=>b, meaning an array with index set a and element type b, in analogy with the type of functions a -> b. (=> associates to the right and binds tighter than ->.)

We can think of Dex as a generalization of the popular einsum DSL for tensor contractions. Here are some examples with their einsum equivalents (written backwards to make the comparison clearer).

```
outer :: i=>Real -> j=>Real -> i=>j=>Real
outer x y = for i j. x.i * y.j
-- 'ij<-i,j' in einsum (written output-first)

matvec :: i=>j=>Real -> j=>Real -> i=>Real
matvec x y = for i. sum (for j. x.i.j * y.j)
-- i<-ij,j

matmul :: i=>k=>Real -> k=>j=>Real -> i=>j=>Real
matmul x y = for i j. sum (for k. x.i.k * y.k.j)
-- ij<-ik,kj

trace :: i=>i=>Real -> Real
trace x = sum (for i. x.i.i)
--  <-ii
```

These functions are all rank monomorphic. To map over extra dimensions we use more index comprehensions. Given a stack of square matrices, z::n=>d=>d=>Real we could obtain the stack of scalar-valued traces as for i. trace z.i.

---

[1]Note that we use '.' for indexing as well as for terminating formal parameter lists.

# 3  Index sets as types

So far, we've skirted around a central question: what does the index i range over in for i. <expr>? The answer is that i ranges over all the values in its type, which must be a finite index set. The type-class IndexSet admits literal sets, tuples of index sets and (constrained) type variables, but not Int, Real, a -> b, or a=>b.

The type system is a very straightforward Hindley-Milner variant. We can choose to explicitly annotate the index binder with its type, as for i::n. <expr>. But we usually let type inference supply the annotation for us. For example, here's our earlier $L_1$ example with the explicit type annotations filled in:

```
pairwiseL1 :: A n d. n=>d=>Real -> n=>n=>Real
pairwiseL1 =
  lam x::(n=>d=>Real) .
    for i::n.
      for j::n.
        sum @d (for k::d. abs (x.i.k - x.j.k))
```

This is a lot closer to a C program. The loops have explicit ranges now, but the ranges are type variables rather than ordinary terms. The scoping rules are similar to Haskell's lexically scoped type variables: n and d are bound by the universal quantifier A. These are actually type arguments to the function, just like the array-length arguments in our C example. We also see an example of type application, sum @d, passing a type argument d to sum :: A n. n=>Real -> Real.

# 4  Structured Index Sets

Is our Hindley-Milner-based type language expressive enough to admit the programs that numerical programmers want to write? What about the dreaded reshape, which would seem to require type-level arithmetic? We hypothesize that the main use of reshape is to group dimensions together, rather than to arbitrarily reinterpret the data buffer. This common case is better served with actual groups—Dex allows product types for index sets for this reason.

As an example, imagine wanting to compute pairwise $L_1$ distances within a batch of images. In NumPy, we would `reshape` each image into a flat vector to use our `pairwiseL1` function:

```
Nbatch, Nx, Ny = images.shape
vecs = np.reshape(images, (Nbatch, Nx * Ny))
dists = pairwiseL1(vecs)
```

In Dex, our `pairwiseL1` function is polymorphic in the index set represented by d, which allows it to be used with a product index set `(Nx,Ny)`. We can rearrange the two spatial indices to be a single index of pairs:[2]

```
-- images :: Nbatch=>Nx=>Ny=>Real
vecs :: Nbatch=>(Nx,Ny)=>Real
vecs.i.(j,k) = images.i.j.k
dists = pairwiseL1 vecs
```

By using pattern matching to group axes, we avoid a whole class of bugs that can arise from getting the shape arithmetic wrong or forgetting how dimensions are ordered [1, 8]. And while type inference with shape arithmetic easily becomes undecidable, type inference for product types is straightforward.

# 5   Dynamic shapes

What about truly data-dependent shapes? The classic example is `filter`, which selects some number of elements of an array based on a predicate. Dex uniformly bails out of these situations using existential types:

```
filter :: (a -> Bool) -> m=>a -> E n. n=>a
```

`filter` returns a standard existential package [6]. At runtime, this carries both a concrete index set n and a value of type `n=>a`, just like arrays in other languages are packaged together at runtime with their sizes.

Putting an existential quantifier on the right-hand side of a table arrow gives us a ragged array: `n=>(E m. m=>Real)`. The inner sizes m are not only statically unknown, but can also vary

---

[2]The transformation from `x` to `x_vec` is analogous to uncurrying a function as `f' (x, y) = f x y`.

based on the index `n`. The other arrays we have seen so far have all been rectangular—in a type like `n=>m=>Real`, the m cannot depend on n because it is bound outside the scope of the table arrow `n=>`. (whether universally or existentially). The type-level distinction between ragged and rectangular arrays lets us smoothly integrate both in the language, while allowing the compiler to emit efficient indexing code in the rectangular case.

# 6   Help wanted

Dex's type system is still a work in progress. At this workshop, we'd like help thinking about limitations such as the following.

## 6.1   Dynamic indexing

Stencil computations are naturally expressed using arithmetic on indices. For example, we might want to write a 1D boxcar blur as:

```
-- Not valid Dex!
blur :: n=>Real -> (n-2)=>Real
blur x = for i. x.(i-1) + x.i + x.(i+1)
```

What does it mean to add an integer to an index? If the index set is a tuple, or a noncontiguous set of integers, it's unclear. Even if the index set is a contiguous set of integers, the new index `i+1` may be out of bounds. Checking this statically requires reasoning about which values the integer offset could take, which could be arbitrarily hard. Then, if we only want to emit the valid subarray, we'd need a type like `blur :: n=>Real ->  (n-2)=>Real`, introducing arithmetic at the type level too!

One proposal, not yet implemented, is to introduce another typeclass constraint on index sets. This would require instances to implement `idxAdd :: n -> Int -> n`, adding an integer to an index, returning an index in the *same* index set. Importantly, this would entail making a choice, encoded in the index set itself, about boundary behavior, for example, wrapping, reflecting, or sticking. Then `blur` would return an array of the same shape as its input.

## 6.2 Other expressiveness limits

How big of a problem is Dex's lack of shape arithmetic in practice? In Section 4 we proposed that most uses of `reshape` should be doable by grouping related dimensions into a tuple-structured index set. Likewise, `concat` corresponds to a type-level sum of index sets. Is that enough? What's a good collection of example programs on which to check whether Dex has enough coverage? Will we need some sort of shape-level casting mechanism to let users escape from the type system? How often?

## 6.3 Efficient ragged arrays

While existential types provide a nice surface language for expressing ragged array computations, our development effort has so far focused on efficiently compiling operations on rectangular arrays. How should ragged arrays be represented internally? How should they be computed on? Pointwise operations seem like a straightforward application of the ideas from nested data parallelism, e.g. [2]; is there a good story for zips and reductions? On accelerators?

## 7 Related work

Index comprehension like Dex's appears in a few other systems. Probably the most prominent in the data science community is the Einstein notation DSL popularized by NumPy's `einsum` and replicated in machine learning frameworks like TensorFlow and PyTorch. The major difference is that einsum only expresses computations that are multilinear in the indexed dimensions.

A Halide [7] or Tensor Comprehensions [12] program specifies the desired computation with a similar explicit indexing notation. Halide and TC also infer iteration ranges, but neither makes the leap to treating index sets as first-class types. This difference gives Dex three advantages: One can represent ragged arrays with existentials (Section 5); one can annotate the type of a shape-polymorphic function and have the Dex compiler check it; and shape polymorphism extends smoothly to groups of dimensions packed in a structured index set (Section 4).

Our index comprehension is also reminiscent of the lazy `build` combinator, a common feature in functional programming languages that enables array fusion [4, 3]. Recently these ideas have been used for the implementation of a differentiable array programming DSL [9], F-smooth. Unlike F-smooth's `build`, the range of Dex's `for` construct is a type, which can be inferred and statically checked. Projecting from the F-smooth work, we anticipate automatic differentiation to be implementable in Dex even as index spaces become richer, and some of their compiler optimizations may be fruitfully applicable as well.

In a different direction, the Remora language [10, 11] is a modern attempt to statically type array programs, with an emphasis on the pointfree broadcasting style instead of explicit indexing. Remora is architected around a cleverly restricted dependent type system that can handle type-level shape arithmetic. The advantage is being able to type a wide range of array programs, including rank polymorphism and `reshape`. A disadvantage is that type checking Remora is complex to describe and implement, and type inference remains an open problem.

Funsors [5] exploit a similar analogy between functions and arrays in probabilistic programming, to uniformly handle continuous and discrete probability distributions.

## 8 Conclusion

We described typed index sets in Dex, a new language for array programming. Dex offers an expressive and lightweight index comprehension syntax, supporting clear and concise array programs. Tracking iteration ranges in the type system exposes them to the compiler, supporting optimizations and parallelism, as well as letting Dex statically detect common shape errors. Treating the index sets as first-class types moreover gets them mostly inferred, and broadens the scope of shape polymorphism to cover common dimension groupings, obviating most of the need for type-level arithmetic.

# 9 Acknowledgements

We thank Martín Abadi, Gilbert Bernstein, James Bradbury, Roy Frostig, Peter Hawkins, Michael Isard, Chris Leary, George Necula, Adam Paszke, Brian Patton, Dan Piponi, Gordon Plotkin, Jonathan Ragan-Kelly, Rif A. Saurous, Olin Shivers, Justin Slepak, Skye Wanderman-Milne, and Alex Wiltschko for helpful conversations.

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
