# OpenReview forum: "Dex: array programming with typed indices"
_NeurIPS.cc/2019/Workshop/Program_Transformations — Program Transformations @NeurIPS2019 Poster_

### Official Review · AnonReviewer1 · 2019-09-26
**Interesting novel syntax for array operations. Good but probably out of scope for this workshop.**

**Confidence:** 4
**Rating:** 6

**Review:**

Proposal for a syntax to express array operations, that should be at the same time clear (unambiguous) and concise. Based on a clever idea of introducing a new sort of *type* for array indices. Many advantages claimed, including simpler analysis and detection of static properties. The proposed syntax is certainly elegant and probably powerful. A stimulating paper, that also proposes new questions and research directions.

Based on the author's reply, I admit that this abstract is more in the scope than I originally thought.  Based on the other abstracts that I saw, and to remain fair, I can slightly increase the rating I chose.

---

### Official Review · AnonReviewer2 · 2019-09-30
**A language and type system for index sets**

**Confidence:** 4
**Rating:** 7

**Review:**

The paper presents a domain-specific array processing language, Dex, that aims to generalize the popular einsum notation in a strongly typed setting. The key innovation is to treat array construction in analogy to lambda, but with index sets as types rather than values. Conventional type inference approaches in the style of Hindley-Milner-Milner then permit inferring iteration spaces. There are a number of interesting aspects about Dex, which will I think lead to good discussions at the workshop. In particular I liked the idea of structured index sets, that appear to elegantly sidestep full-blown type-level arithmetic while still supporting large classes of use cases for highly polymorphic operations like reshape. The paper does a good job outlining not only strengths but also limitations, e.g. around dynamic indexing or ragged layouts.

---

### Decision · Program_Chairs · 2019-10-01

**Decision:**

Accept (Poster)

**Comment:**

The reviewers thought this was a strong contribution. Although Dex is a language that could be easily amenable to program transformations, it is itself not a program transformation nor a program transformation framework, and as such it is perhaps not squarely in the scope of the workshop.